# Comprehensive Analysis of the *UGT* Gene Superfamily in *Spodoptera frugiperda*

**DOI:** 10.3390/insects16060601

**Published:** 2025-06-06

**Authors:** Yang Liu, Minghui Guan, Kunliang Zou, Tonghan Wang, Haiyang Wang, Lu Sun, Bo Feng, Jiali Ding, Xiang Gao, Yongfu Wang, Degong Wu, Junli Du

**Affiliations:** 1College of Resources and Environment, Anhui Science and Technology University, Chuzhou 233100, China; 18298115131@163.com (Y.L.); mhbelucky@163.com (M.G.); sunluslu@163.com (L.S.); 2College of Agriculture, Anhui Science and Technology University, Chuzhou 233100, China; zkl151231@163.com (K.Z.); 18767401521@163.com (T.W.); 17334571480@163.com (H.W.); 18110514508@163.com (B.F.); 18130751909@163.com (J.D.); 19565987398@163.com (X.G.); wangyongfu@ahstu.edu.cn (Y.W.)

**Keywords:** *Spodoptera frugiperda*, UDP-glycosyltransferase (UGT), gene family analysis

## Abstract

The fall armyworm is a major pest that causes significant damage to crops. To better understand how this insect copes with harmful substances, we investigated a group of genes known as UDP-glycosyltransferases (UGTs), which play key roles in toxin removal. Through computational analysis, we identified 48 *UGT* genes in the fall armyworm, some of which appear to have expanded through gene duplication. These genes predominantly encode proteins characterized by α-helical secondary structures. We also observed variations in gene organization and protein sequences. Comparative analysis with other insects revealed that the fall armyworm is closely related to Spodoptera litura. Several *UGT* genes showed stage-specific and tissue-specific expression patterns during larval development, indicating their involvement in the digestion, detoxification, and transport of important molecules. This study enhances our understanding of the fall armyworm’s survival mechanisms and could contribute to developing new strategies for pest management.

## 1. Introduction

The fall armyworm (*Spodoptera frugiperda*) is a destructive migratory pest native to the tropical and subtropical regions of the Americas [1], causing severe damage to agricultural crops, especially maize [2]. Studies have documented that *S. frugiperda* infestations can reduce crop yields by over 50% under severe conditions [3]. During the late instar stage, the larvae’s voracious feeding behavior leads to extensive damage to a variety of crops, including maize, rice, and others, by severing leaf veins and disrupting nutrient transport [4]. The invasive spread of the pest has caused significant declines in maize production in Honduras, with yield losses in Brazil and Argentina ranging from 34% to 72% [5]. In South Africa, annual maize production has plummeted by 16.1 million tons, resulting in an economic loss of up to CNY 6.3 billion [6]. *S. frugiperda*’s strong adaptability allows it to survive in a broad range of environments, except in cold regions where low temperatures are detrimental to its survival. The pest’s distribution now spans from southern California to Argentina, posing a significant threat to global agricultural production [7].

*UGT* enzymes, involved in glycosylation reactions, constitute the largest family of glycosyltransferases in plants and are also widely present in animals, bacteria, and viruses [8]. They catalyze the transfer of glycosyl groups, affecting the solubility, stability, and biological activity of acceptor molecules, thereby altering their properties [9]. This process converts hydrophobic substances into water-soluble forms [10], enhancing their solubility, reducing toxicity, and facilitating excretion [11]. In nature, the feeding interactions between insects and plants drive the evolution of various resistance mechanisms in plants, while insects develop physiological and biochemical adaptations to mitigate the adverse effects of ingesting plant toxic compounds [12]. Through glycosylation reactions catalyzed by *UGT* enzymes, insects can degrade toxic substances in their food, thereby reducing the harmful effects of these compounds [13]. Glycosylation plays a crucial role in detoxification processes within organisms, aiding in the elimination and suppression of both endogenous and exogenous toxic substances [14].

The UGTs in insects are essential for numerous physiological processes, such as the detoxification of secondary metabolites from plants, the formation of the cuticle, the development of pigmentation, and the functioning of the olfactory system [15]. Pym et al. confirmed through transcriptome analysis that *UGT344P2* is highly overexpressed in resistant clones, indicating that it significantly confers resistance to sulfoxaflor [16]. Zheng et al. discovered, via transcriptomics and proteomics, that the expression of *UGT* genes in *Helicoverpa armigera* is upregulated after ingesting gossypol and tannins from cotton, suggesting their involvement in detoxification and digestion [17]. Yan et al.’s research demonstrated that *UGT* genes in parasitoid wasps provide detoxification effects against certain concentrations of phoxim, cyfluthrin, and chlorfenapyr; when three *UGT* genes were knocked out, the mortality rate of the wasps significantly increased under sublethal doses of insecticides [18]. Ahn et al. revealed that *UGT* enzymes catalyze the conjugation of various lipophilic small molecules with sugars to form glycosides, playing a key role in detoxifying xenobiotics and regulating endogenous metabolites in insects [19].

*UGT* enzymes are also critical for the detoxification processes in the fall armyworm [20]. For example, by using dsRNA interference targeting the *SfUGT33F28* gene, specific UDP-glycosyltransferases can be inhibited, thereby deactivating the defensive toxic substances in maize. This highlights the significant role of UGTs in glycosylation within *S. frugiperda* [21]. Therefore, further investigation into *UGT* enzymes in *S. frugiperda* is essential for elucidating its detoxification mechanisms [22].

A bioinformatics analysis was conducted on the *SfUGT* gene family, examining its expression patterns across different developmental stages and tissues of *S. frugiperda*. Analyzing the functions of these genes identifies critical targets for upcoming gene knockout experiments and RNAi-mediated approaches aimed at pest management. Additionally, these results present important perspectives for creating innovative pesticides.

## 2. Materials and Methods

### 2.1. Identification and Acquisition of the SfUGT Gene Family

The genome sequence and annotation information for *S. frugiperda* were retrieved from the NCBI (https://www.ncbi.nlm.nih.gov/, accessed on 20 May 2024) database and the genome version provided by Zhejiang University. *UGT*-domain-containing homologous protein sequences were identified using HMMER (version 3.3.2) with the Pfam (http://pfam.xfam.org/, accessed on 25 May 2024) *UGT* domain model (PF00201) as query, applying an E-value cutoff of 1 × 10^−5^. The sequences were further verified by SMART (Version 9.0) and NCBI CDD (https://www.ncbi.nlm.nih.gov/Structure/cdd/wrpsb.cgi, accessed on 27 May 2024) to exclude false positives. The filtered *UGT* candidates were then localized using TBtools [23] (v2.096) genome annotation tools following Liu et al. [24].

### 2.2. Analysis of Gene Duplication and Ka/Ks Ratios for SfUGT

The One Step MCScanX plugin in TBtools [23] was utilized to analyze gene duplication relationships. Subsequently, the associated genes were imported into the Simple Ka/Ks Calculator plugin to calculate the rates of synonymous (Ks) and nonsynonymous (Ka) substitutions. The divergence time was calculated using the formula T = Ks/(2λ) (λ = 6.5 × 10^−9^) [25].

### 2.3. Physicochemical Property Analysis of SfUGT Proteins

The physicochemical properties of SfUGT proteins were predicted using the ProtParam tool on ExPASy [26] (https://web.expasy.org/protparam/, accessed on 6 June 2024). Subcellular localization was analyzed using WOLFPSORT [27] (https://wolfpsort.hgc.jp/, accessed on 10 June 2024), and protein secondary structures were predicted and analyzed using SOPMA [28] (https://npsa-pbil.ibcp.fr/cgi-bin/npsa_automat.pl?page=/NPSA/npsa_sopma.html, accessed on 20 June 2024).

### 2.4. Structural Characterization and Motif Landscape of SfUGT Genes

MEME [29] (Version 5.5.5) (https://meme-suite.org/meme/tools/meme, accessed on 7 July 2024) was used to analyze motifs in the proteins of *S. frugiperda* (a total of 10 motifs were identified, while other settings were maintained at default values). Subsequently, gene structure, conserved motifs, and phylogenetic trees of the *UGT* family were integrated and analyzed using TBtools [23].

### 2.5. Phylogenetic Analysis of SfUGT Across Species

To investigate the evolutionary relationships between *S. frugiperda* and other species, genome sequences and annotation files for *S. litura* (ASM270686v3) and *Helicoverpa armigera* (ASM3070526v1) were retrieved from the NCBI database. Homologous sequences containing *UGT* domains were identified using the method described by Liu Yang et al. [24]. Multiple sequence alignment of the protein sequences was performed using MAFFT (v7.505), and the alignment quality was assessed with Guidance2. The phylogenetic tree was constructed using IQ-TREE (v2.1.2), with ModelFinder employed to select the best-fit evolutionary model. Node support was evaluated by 1000 ultrafast bootstrap replicates and SH-aLRT tests. The resulting phylogenetic tree was visualized using iTOL (v6).

### 2.6. Expression Analysis of the SfUGT Gene Family at Different Developmental Stages

Transcriptome data with accession numbers PRJNA590312 [30] and PRJNA1070356 [31] were downloaded from the NCBI SRA database (https://www.ncbi.nlm.nih.gov/sra, accessed on 15 July 2024). Heatmaps of *SfUGT* gene family expression levels were generated using the HeatMap tool in TBtools [23] to process data from different developmental stages and tissues.

### 2.7. qPCR Analysis of the SfUGT Gene

Insect Rearing

*S. frugiperda* larvae were originally collected from maize experimental fields at Anhui Science and Technology University in July 2024. The maize fields were managed under typical local agronomic conditions without pesticide application prior to collection. The collected larvae were reared for no more than three successive generations under laboratory conditions before being used for subsequent experimental analyses to minimize potential effects of laboratory adaptation.

Insects were maintained in a climate-controlled chamber at (25 ± 1) °C, 70 ± 5% relative humidity, and a photoperiod of 16 h light and 8 h darkness (16:8, L:D). Larvae were fed fresh maize leaves, and adults received a supplement of 10% honey water.

Samples were collected from various developmental stages of *S. frugiperda*: eggs (*n* = 80), first instar larvae (*n* = 10), second to sixth instar larvae (*n* = 5 each), pupae (*n* = 5), and various tissues (fat body, Malpighian tubules, midgut, head, and hemolymph) from sixth instar larvae, with three biological replicates per sample. All samples were rapidly snap-frozen in liquid nitrogen and stored at −80 °C until RNA extraction.

RNA was extracted, and reverse transcription was carried out according to the protocol described by Lv et al. [32]. Primers were designed using the Primer 3.0 [33] tool (Table 1), with GAPDH serving as the reference gene due to its stable expression across samples. Primer specificity was confirmed by melt curve analysis and agarose gel electrophoresis, ensuring single specific amplification products. Real-time quantitative PCR (qPCR) was performed according to the method outlined by Jin et al. [34] to assess the expression levels of seven *SfUGT* genes across various developmental stages and tissues. Each sample included three biological replicates, with each biological replicate containing three technical replicates. Relative expression levels were calculated using the 2^−ΔΔCT^ method [35]. PCR amplification efficiencies for each primer pair were evaluated and found to be within the acceptable range (90–110%). Data are presented as mean ± standard error, organized and analyzed using SPSS 28 [36] and visualized with GraphPad Prism 9.5.0 [37]_._ In this study, we focused on describing gene expression patterns and did not perform statistical significance tests.

## 3. Results

### 3.1. Discovery and Characterization of the SfUGT Gene Family

A total of 48 genes from the *UGT* family were identified in the *S. frugiperda* genome database, and they were sequentially named based on their chromosomal positions (Figure 1). The 48 *UGT* genes were unevenly distributed across 10 chromosomes. Interestingly, chromosome Chr25 harbored the highest number of *UGT* genes (16), followed by Chr27 with 14, Chr7 with 5, and Chr12 with 4. Chr10, Chr26, and NW_023337172.1 each contain 2 genes, while the remaining chromosomes had only 1 gene. Interestingly, *SfUGT47* and *SfUGT48* are both located on chromosomal fragments.

### 3.2. Analysis of SfUGT Gene Duplication and Ka/Ks Ratios

By analyzing the evolutionary trends and chromosomal positions of the gene family members, 23 tandemly duplicated gene pairs were identified (Table 2), indicating that *UGT* gene family expansion events were relatively frequent. As shown in Table 2, the Ks values of the duplicated gene pairs exceed the Ka values, indicating a higher frequency of synonymous substitutions. All duplicated gene pairs exhibited Ka/Ks ratios below 1, indicating they were subject to purifying selection with limited functional divergence. According to Table 2, it is evident that the Ks values of the gene pairs that have duplicates exceed the Ka values, implying a greater prevalence of synonymous substitutions. Furthermore, the Ka/Ks ratios for all duplicated gene pairs were under 1, indicating strong purifying selection and limited functional divergence.

The occurrence of NaN values for the Ka/Ks ratios in two gene pairs is primarily due to excessive sequence divergence, which hinders the accurate estimation of substitution rates. Biologically, this typically indicates that the gene pairs have a very distant evolutionary relationship, having accumulated numerous mutations. In particular, saturation at synonymous sites can cause the Ks values to approach zero or become unstable, resulting in anomalous Ka/Ks calculations. Table 2 shows the *SfUGT* gene replication and Ka/Ks value analysis in *S. frugiperda*.

### 3.3. Construction of SfUGT Phylogenetic Tree and Protein Structure Analysis

#### 3.3.1. Analysis of SfUGT Protein Structure

By analyzing the physicochemical properties of the SfUGT protein genes, we can directly infer certain functional characteristics of the organism. Based on the analysis (Table 3), the number of amino acids in the proteins ranges from 236aa to 1102aa. Except for a few proteins—such as SfUGT1, SfUGT2, SfUGT30, SfUGT37, and SfUGT38, which have relatively fewer amino acids, and SfUGT28 and SfUGT48, which have relatively more—the differences in amino acid counts are not significant, with most proteins containing around 500 amino acids. The molecular weights of these proteins vary from 32,873.72 Da to 124,725.23 Da. Based on the provided data, there appears to be a clear positive correlation between the number of amino acids and molecular weight. The isoelectric points (pI) range from 5.89 to 9.37, with the majority (39 proteins) being classified as acidic proteins, while only a few (8 proteins) are basic proteins. Additionally, one protein, SfUGT16, is neutral.

It is generally accepted that a protein is considered stable if its instability index is below 40. In this study, the instability index of proteins, such as SfUGT5, SfUGT15, SfUGT16, SfUGT17, SfUGT18, SfUGT22, SfUGT28, SfUGT30, SfUGT36, SfUGT37, and SfUGT38, is below 40, indicating they are stable proteins, while the other 37 proteins are classified as unstable. Based on the aliphatic index data, the majority of the 48 SfUGT proteins have an index around 100. Analysis of the grand average of hydropathicity (GRAVY) shows that 31 of these proteins have negative values, suggesting they are likely hydrophilic, while the remaining 17 proteins, with positive GRAVY values, are predicted to be hydrophobic.

The predicted subcellular localization data indicate that 26 proteins are localized to the endoplasmic reticulum, 14 to the plasma membrane, 3 to the extracellular space, 2 to peroxisomes, and 1 protein each in the cytoplasm, mitochondria, and nucleus. These results reveal a specific distribution pattern of SfUGT proteins, suggesting they may perform multiple functions.

#### 3.3.2. Protein Secondary Structure Analysis

The function of a protein is dictated by its composition. According to the ratio of α-helix, β-turns, random coils, and extended strands (Table 4), the α-helix accounts for a relatively high percentage, primarily around 45%. In contrast, the β-turns show a relatively low proportion, generally below 10%. It is speculated that the α-helix structure plays a key role in the secondary structure of SfUGT proteins, while random coils and extended strands may serve secondary roles, and β-turns likely function as structural modifiers.

### 3.4. Analysis of Phylogenetic Relationships, Gene Architecture, and Conserved Motifs in SfUGT Genes

A phylogenetic tree without roots was created using the multiple sequence alignments of the *SfUGT* genes (refer to Figure 2). The phylogenetic tree’s classification reveals that the 48 genes can be categorized into three separate groups. Group B has the highest representation, containing a total of 23 genes. In contrast, Group A includes 15 genes, whereas Group C is the smallest group, consisting of only 10 genes.

As shown in Figure 2, the number and length of introns and UTRs in the 48 *SfUGT* genes vary to some extent. Except for *SfUGT30*, *SfUGT37*, and *SfUGT38* in Group B, all the other genes contain both CDS and UTR regions.

The analysis of conserved motifs revealed that *SfUGT* genes possess 10 distinct conserved motifs, with relatively small variations in motif length. This indicates a high level of conservation among the gene members within the *UGT* family. A study on the distribution of conserved motifs showed that *SfUGT1*, *SfUGT2*, and *SfUGT30* exhibit significant motif loss, retaining only four conserved motifs. This suggests that the functions of *SfUGT1*, *SfUGT2*, and *SfUGT30* may have diverged.

Further analysis of the conserved domains of *SfUGT* genes revealed that *SfUGT3*, *SfUGT6*, *SfUGT12*, *SfUGT13*, *SfUGT19*, *SfUGT22*, *SfUGT31*, and *SfUGT32* possess a GT1_Gtf-like domain, while most of the remaining genes belong to the Glycosyltransferase_GTB-type superfamily.

### 3.5. Phylogenetic Analysis of SfUGT Proteins Across Species

As shown in Figure 3, the *UGT* genes are divided into three groups. Group A contains the fewest *UGT* genes, with 30 genes, while Group C has the largest number, with 65 genes. Group B includes 57 genes.

In this study, 58 pairs of homologous relationships were identified among the *UGT* genes, accounting for approximately 76.31% of the total genes. Across different species, 30 pairs of orthologous genes were found, with 27 pairs between *S. frugiperda* and *S. litura* and 3 pairs between *H. armigera* and *S. litura*. No direct orthologous relationships were observed between *S. frugiperda* and *H. armigera*. The high number of orthologous genes shared between *S. frugiperda* and *S. litura* suggests a closer evolutionary relationship, while the lack of direct homologs between *S. frugiperda* and *H. armigera* indicates a more distant phylogenetic connection.

### 3.6. Analysis of SfUGT Gene Family Expression Across Different Developmental Stages

According to the expression profiles observed at different developmental stages (Figure 4), it was found that genes *SfUGT15*, *SfUGT30*, *SfUGT32*, *SfUGT35*, and *SfUGT48* exhibited higher expression levels across all growth stages compared to other genes. In contrast, the expression levels of *SfUGT1*, *SfUGT18*, and *SfUGT29* were relatively low throughout the developmental stages, with their TPM values being below 1. The expression levels of *SfUGT9* and *SfUGT23* were higher from the first to the sixth instar larval stages, as well as in the adult female and male stages. Additionally, *SfUGT20*, *SfUGT24*, and *SfUGT40* showed relatively higher expression levels from the first to the fifth instar stages.

### 3.7. Tissue-Specific Expression Analysis of the SfUGT Gene Family

From the heatmap of the expression profiles across various tissues (Figure 5), it can be observed that genes *SfUGT9*, *SfUGT15*, *SfUGT17*, *SfUGT20*, *SfUGT23*, *SfUGT24*, *SfUGT28*, *SfUGT30*, *SfUGT32*, *SfUGT33*, *SfUGT39*, *SfUGT40*, *SfUGT42*, *SfUGT43*, and *SfUGT45* were highly expressed in the Malpighian tubules. Genes *SfUGT32*, *SfUGT35*, *SfUGT36*, and *SfUGT42* were more highly expressed in the head, while *SfUGT14*, *SfUGT15*, *SfUGT20*, *SfUGT35*, *SfUGT36*, *SfUGT37*, and *SfUGT48* had the highest expression levels in the fat body. In the midgut, genes *SfUGT6*, *SfUGT9*, *SfUGT24*, *SfUGT40*, and *SfUGT42* were more highly expressed. Furthermore, *SfUGT21*, *SfUGT26*, *SfUGT30*, *SfUGT39*, *SfUGT40*, and *SfUGT48* were more highly expressed in the hemolymph.

### 3.8. qPCR Analysis of SfUGT Gene Expression Across Developmental Stages

The expression patterns of *SfUGT* genes at different developmental stages of *S. frugiperda* were analyzed using qPCR (Figure 6). The results revealed that the expression level of *SfUGT9* was downregulated during the sixth instar but increased in the first through fifth instars. *SfUGT16* showed a significant upregulation during the first instar, with lower expression levels in other developmental stages, suggesting that *SfUGT16* may be specifically upregulated during the first instar. *SfUGT21* exhibited a notable upregulation during the fourth instar, while its expression was low in both male and female adults. *SfUGT30*’s expression was elevated across all developmental stages, with the highest levels observed during the first instar. *SfUGT34* and *SfUGT40* had the highest expression during the third instar but showed lower expression from the pupal stage to the adult stage. Notably, *SfUGT34* expression was downregulated during the first instar, while *SfUGT40* exhibited the opposite trend. *SfUGT48* expression increased in each developmental stage, peaking during the pupal stage.

### 3.9. qPCR Analysis of SfUGT Gene Expression in Different Tissues

The tissue-specific expression patterns of *UGT* genes in *S. frugiperda* were examined using qPCR (Figure 7). The results showed that *SfUGT9* exhibited the highest expression in the Malpighian tubules, while its expression in other tissues was relatively low, indicating a potential tissue-specific high expression of *SfUGT9* in the Malpighian tubules. The expression of *SfUGT16* was higher in both the midgut and Malpighian tubules, while expression was downregulated in other tissues. *SfUGT21* and *SfUGT30* were significantly upregulated in the Malpighian tubules and hemolymph, but *SfUGT21* was upregulated in the fat body and downregulated in the head, whereas *SfUGT30* showed the opposite pattern. *SfUGT34* was notably upregulated in the Malpighian tubules, followed by the head. *SfUGT40* exhibited higher expression in the midgut and hemolymph, with lower expression in other tissues. Finally, *SfUGT48* was markedly upregulated in the hemolymph, followed by the head.

## 4. Discussion

Gene duplication serves as the foundation for the functional diversification of homologous genes and is the primary mechanism generating new functional genes. It acts as a key driver of genome and species evolution [38]. In the genome of *S. frugiperda*, 48 *UGT* genes were identified and unevenly distributed across 10 chromosomes, with 23 tandem duplication events. In contrast, Mei Yang et al. only identified 39 *UGT* genes in their evolutionary analysis of detoxification metabolism-related gene families in *S. frugiperda* [39]. This study demonstrates that tandem duplication is the primary mechanism underlying the expansion of the *SfUGT* gene family.

The secondary structures of these proteins are predominantly composed of α-helices. Analysis of gene structure and conserved motifs revealed notable differences in intron numbers among *SfUGT* genes, while the diversity in amino acid sequences suggests potential functional divergence among the encoded enzymes. This suggests that *UGT* enzymes possess high catalytic activity and a notable detoxification effect. The *UGT* gene family was categorized into four subfamilies by the phylogenetic tree, demonstrating a significant presence of orthologous genes in *S. frugiperda* and *S. litura*, which suggests a close evolutionary connection [40]. This finding implies that *S. frugiperda* and *S. litura* may have evolved comparable ecological adaptations.

Ahn et al. [19] illustrated that UGTs facilitate the conjugation of different lipophilic small molecules with sugars, resulting in the formation of glycosides. This process is thought to contribute to the detoxification of xenobiotics and regulation of endogenous substances in insects, though direct biochemical evidence is still required. Many *UGT* genes are expressed in the fat body, midgut, and Malpighian tubules, suggesting potential involvement in detoxification processes. Certain *UGT* genes in *Bombyx mori* are expressed across various tissues and developmental phases, indicating that these genes might have significant roles in the growth and developmental processes of *B. mori* [41].

Expression analysis of *S. frugiperda* at different developmental stages revealed that the relative expression levels of *SfUGT9*, *SfUGT40*, and *SfUGT48* were higher in the first to fifth instar stages, but their expression began to downregulate in the sixth instar. *SfUGT16* and *SfUGT21* were specifically highly expressed in the first and fourth instars, respectively, while *SfUGT34* was upregulated in both the third and fourth instars. In *Spodoptera littoralis*, it was first confirmed that odor exposure modulates *UGT* expression, highlighting *UGT*’s role in olfaction [42]. During the adult stage, female moths need to release large amounts of sex pheromones to attract males for reproduction, and *SfUGT30* is highly expressed in male moths, which may reflect a role related to sex pheromone recognition, although further functional validation is needed [43].

In tissue expression profiles, *SfUGT9* and *SfUGT34* were specifically highly expressed in the Malpighian tubules, suggesting a potential role in digestive metabolism. Studies have shown that *UGT* genes are specifically expressed in the midgut and fat body, potentially related to detoxification functions by neutralizing plant-derived toxins in insect diets [44]. *SfUGT16* was highly expressed in the Malpighian tubules and midgut of *S. frugiperda*, indicating that this gene may be involved in digestive metabolism and detoxification. Insects’ hemolymph, a mixture of blood and lymphatic fluid, circulates in their open circulatory system and plays an essential role in metabolism, serving as a site for storing and transporting various substances. *SfUGT21*, *SfUGT30*, and *SfUGT48* were highly expressed in the hemolymph, suggesting their involvement in metabolic and transport processes. Additionally, *SfUGT40* was highly expressed in both the midgut and hemolymph, indicating potential roles in detoxification and metabolism.

According to other studies, the *UGT* activity in the fat body is highest in the Asian corn borer (*Ostrinia furnacalis*) [45], and *UGT* activity in the midgut is highest in *S. litura* [46]. However, in this study, based on transcriptome data and qPCR analysis, *UGT* activity was highest in the Malpighian tubules of *S. frugiperda*, which might be related to differences among insect species.

Furthermore, the role of the *UGT* gene family in insecticide resistance has received increasing attention. Previous studies have demonstrated that UGTs enhance the metabolic capacity of pests toward chemical insecticides by promoting glycosylation modifications of pesticides and their metabolites, thereby increasing resistance levels [47]. In this study, several *SfUGT* genes exhibited high expression across multiple larval developmental stages and key detoxification organs, suggesting their involvement in the detoxification of plant secondary metabolites and exogenous compounds, which may consequently influence the insect’s resistance phenotype [19]. Future research combining RNAi or gene-editing technologies such as CRISPR/Cas9 to target and knock down these highly expressed *UGT* genes will help elucidate their specific functions and provide a theoretical basis for environmentally friendly pest management strategies [48,49]. Additionally, this study was limited to transcriptional-level analyses, lacking protein function validation and in vivo functional assays. Subsequent work should include protein expression and enzymatic activity assays, as well as gene knockout or overexpression experiments, to further clarify the molecular mechanisms by which *UGT* genes mediate detoxification and insecticide resistance in *S. frugiperda* [50]. Finally, combined with evolutionary analyses, the expansion of the *UGT* gene family may provide the genetic foundation for *S. frugiperda*’s adaptation to diverse host plants and complex environments, offering new insights into its rapid spread and ecological adaptability and identifying molecular targets for the development of more effective control measures [51].

This research examines the *UGT* gene family in *S. frugiperda*, identifying its members and exploring their physicochemical characteristics, gene architecture, subcellular distribution, secondary protein structure, sequence traits, phylogenetic connections, chromosomal positioning, and patterns of gene expression. The analysis of the *UGT* gene family in *S. frugiperda* partially fills a gap in the research on *UGT* genes in this species. Additionally, high-expression genes were identified to explore their potential functions, providing a theoretical basis for future gene knockout techniques and pest control strategies for *S. frugiperda*. The findings may also serve as references for the development of new pesticides.

## 5. Conclusions

In this study, we conducted a comprehensive bioinformatics analysis of the *UGT* gene superfamily in *S. frugiperda*, uncovering several key insights.

1. A total of 48 members of the *UGT* gene family were found, demonstrating a non-uniform distribution among 10 chromosomes, along with the observation of 23 tandem duplication events.

2. The predicted secondary structures of SfUGT proteins were dominated by α-helices, accounting for a major portion of the structural composition.

3. Structural analysis of gene models and conserved motifs revealed intron number variation and amino acid sequence diversity among *SfUGT* genes, reflecting the structural complexity of this gene family.

4. Phylogenetic reconstruction grouped the *SfUGT* genes into four major subfamilies. A greater number of orthologs were found between *S. frugiperda* and *S. litura* compared to other species included in the analysis.

5. Developmental expression profiling by qPCR showed that *SfUGT9*, *SfUGT40*, and *SfUGT48* exhibited higher expression from the first to fifth instar larvae, with a decline in the sixth instar. *SfUGT16* and *SfUGT21* displayed peak expression at the first and fourth instar, respectively, while *SfUGT30* showed its highest expression in the male adult stage.

6. Tissue-specific expression analysis indicated that *SfUGT16* was predominantly expressed in the Malpighian tubules and midgut. High transcript levels of *SfUGT21*, *SfUGT30*, and *SfUGT48* were observed in the hemolymph, while *SfUGT40* was expressed at elevated levels in both the midgut and hemolymph.

In summary, this study fills a critical knowledge gap by characterizing the expression patterns of the *UGT* gene family in *S. frugiperda* across developmental stages and tissues. The identified *UGT* genes serve as promising candidates for future functional studies, including gene knockout and RNAi-based pest control methods. While these findings provide a foundational basis for exploring UGTs as potential targets for pesticide development, direct pesticide response assays are still needed to validate their applicability. Therefore, the implication of these genes in pesticide development remains a prospective avenue for further investigation.

This study validated the expression of target genes using both quantitative PCR and transcriptome sequencing. However, inherent differences in sensitivity, quantification principles, and normalization methods between these platforms led to some discrepancies in gene expression patterns. Additionally, the qPCR results were not analyzed statistically, as the focus was primarily on describing expression trends rather than testing for significance. The evolutionary analysis of Ka/Ks ratios did not include confidence intervals or tests for rate heterogeneity. Moreover, conserved motifs were presented visually without quantitative evaluation. Future studies should employ more rigorous evolutionary models and incorporate quantitative assessments to improve the scientific rigor and interpretability of the results.

## Figures and Tables

**Figure 1 insects-16-00601-f001:**
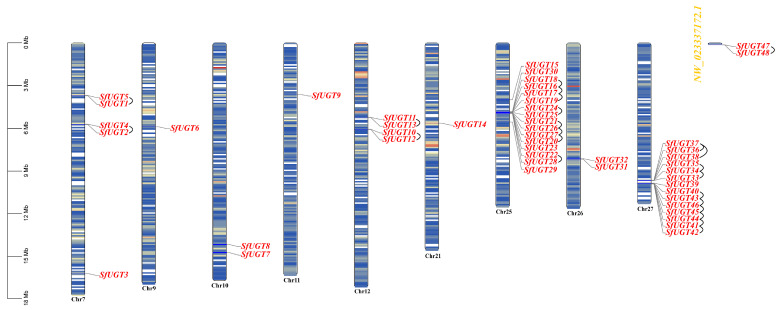
Analysis of chromosomal distribution and gene duplication of the *SfUGT* gene in *S. frugiperda*. Note: To more clearly illustrate the genomic distribution of the genes, this study employed a nomenclature system for *UGT* genes based on their chromosomal physical positions. This approach facilitates genomic structural analysis and functional comparisons, thereby aiding interpretation and application in subsequent research.

**Figure 2 insects-16-00601-f002:**
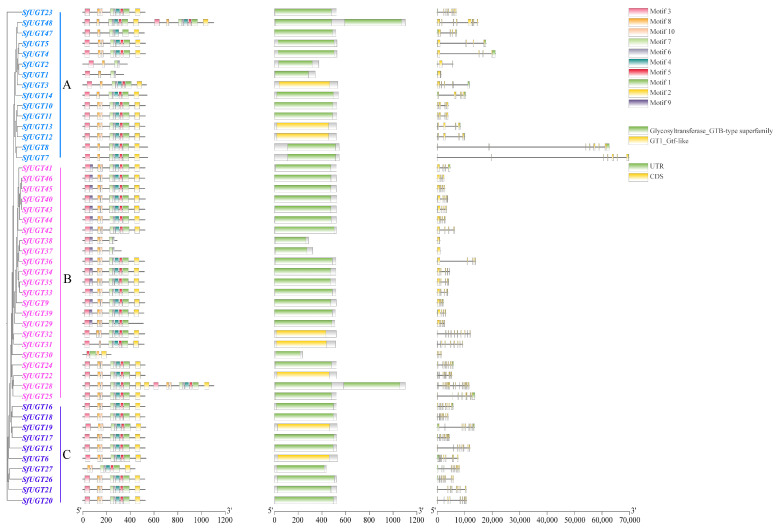
Phylogenetic diagram, gene structure, and investigation of conserved domains within the *SfUGT* family in *S. frugiperda*. Note: The protein functional domain analysis shown on the far right of the figure indicates that most *SfUGT* proteins contain the green-labeled Glycosyltransferase_GTB-type domain, belonging to the GT-B-type glycosyltransferase superfamily. Some proteins also possess the yellow-labeled GT1_Gtf-like domain, which may be involved in specific substrate recognition and glycosyl transfer functions. These domains reflect the core functional features of the *SfUGT* family as glycosyltransferases.

**Figure 3 insects-16-00601-f003:**
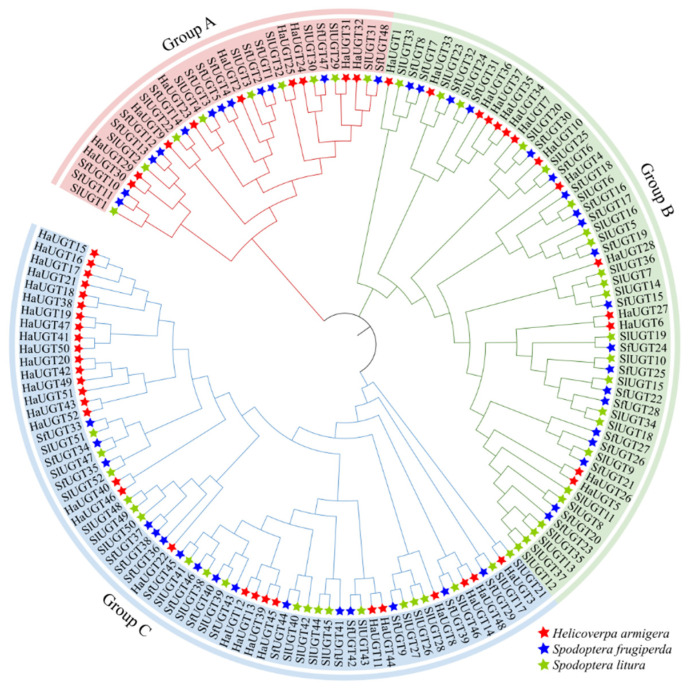
Phylogenetic analysis tree of *SfUGT* family within *Spodoptera frugiperda*, *Spodoptera litura*, *Helicoverpa armigera*. Note: Group A comprises 30 genes, Group B includes 57 genes, and Group C contains 65 genes, representing the largest group. Orthologous genes refer to genes in different species that originated from a common ancestral gene and generally retain conserved functions. These genes typically preserve similar functional roles following species divergence.

**Figure 4 insects-16-00601-f004:**
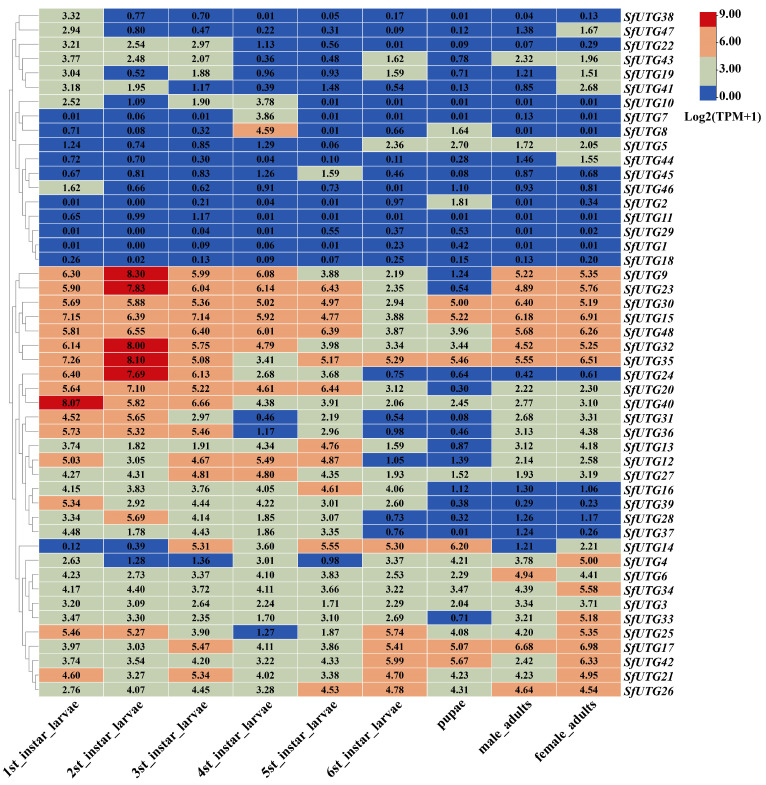
Expression patterns of *SfUGT* at various developmental stages of *S. frugiperda.* Note: The figure presents a heatmap illustrating the expression levels of the *UGT* gene family in *S. frugiperda* across different developmental stages, represented as Log2(TPM+1) values. Higher Log2(TPM+1) values indicate greater gene expression levels. This visualization enables the comparison of temporal expression patterns, highlighting genes that may play key roles at specific growth phases.

**Figure 5 insects-16-00601-f005:**
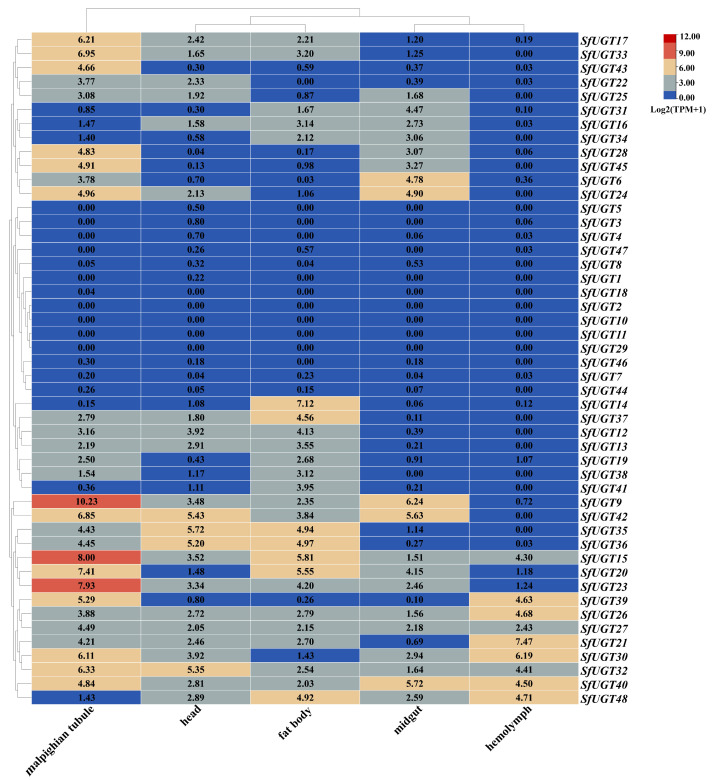
Expression of *SfUGT* in different organs of *S. frugiperda*. Note: This figure presents a heatmap illustrating the expression profiles of the *UGT* gene family in *S. frugiperda* across various tissues, quantified using Log2(TPM+1) values, where higher values indicate greater expression levels. The pronounced differential expression of these genes among tissues provides valuable insights into their tissue-specific functions and underlying biological roles.

**Figure 6 insects-16-00601-f006:**
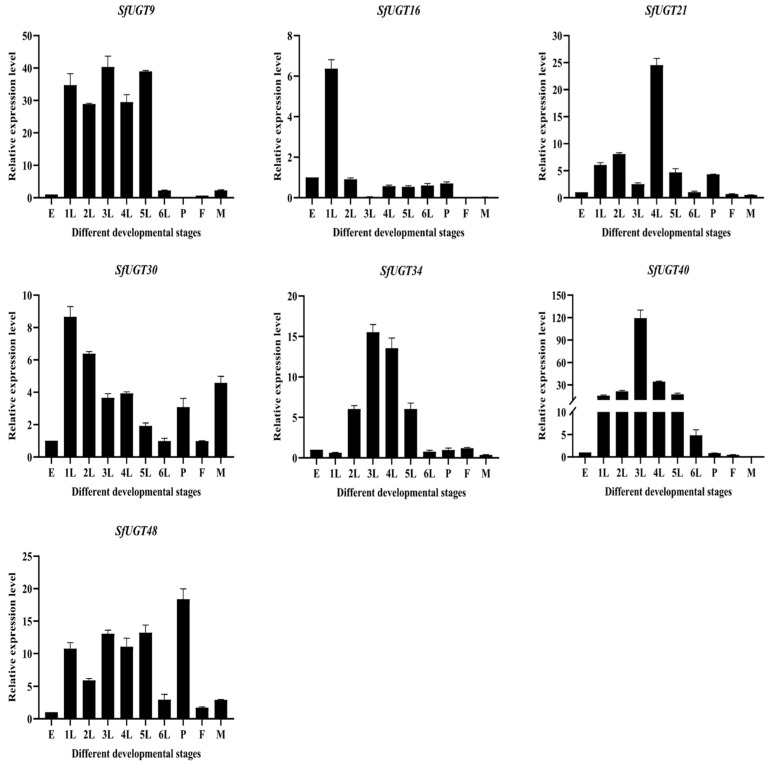
Developmental stage-specific expression patterns of *SfUGT* genes. Note: 1L represents the first instar; 2L, the second instar; 3L, the third instar; 4L, the fourth instar; 5L, the fifth instar; 6L, the sixth instar; P, the pupal stage; F, the adult female stage; and M, the adult male stage. Data are presented as mean values ± standard deviation. The egg stage was used as the reference group for evaluating the expression of seven *SfUGT* genes across different developmental stages of *S. frugiperda*.

**Figure 7 insects-16-00601-f007:**
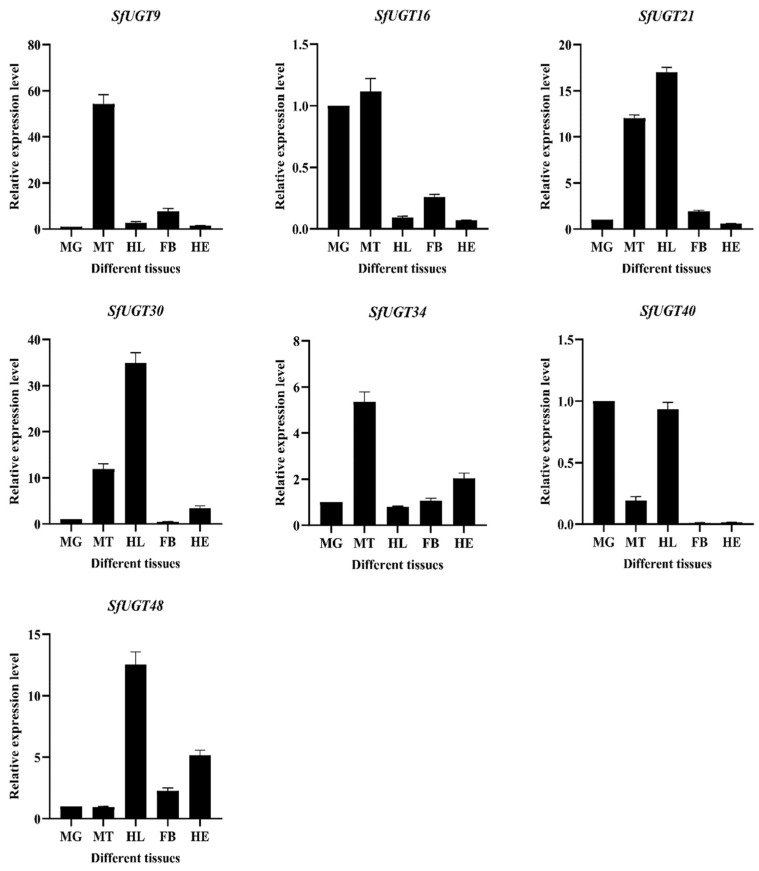
Expression pattern of *UGT* gene in different tissues of *S. frugiperda*. Note: MG: midgut MT: Malpighian tube HL: hemolymph FB: fat body HE: head. In our study of 7 *SfUGT* genes expressed across various tissues in *S. frugiperda*, the midgut was utilized as the control group. The data points reflect the mean ± standard deviation.

**Table 1 insects-16-00601-t001:** List of primers utilized in the current study.

Primer Name	Sequence of Primers (5′-3′)
*GADPH*-*F*	CGGTGTCTTCACAACCACAG
*GADPH*-*R*	TTGACACCAACGACGAACAT
*SfUGT9*-F	ACAATGTCGGTGCTCTGGTT
*SfUGT9*-R	TCTGCGGTGATAACGGTGAC
*SfUGT16*-F	AACTATGGACACGGATTCAATGG
*SfUGT16*-R	CGTACAGCATCATGTTCATTAGC
*SfUGT21*-F	AGGTCCGATACAACGCAACT
*SfUGT21*-R	CAGGTAATGGCTTGACTTCTTCC
*SfUGT30*-F	TGAAGGACTGTGGAAGGTTGT
*SfUGT30*-R	CGGATGAGCGTTAATCAGCATA
*SfUGT34*-F	ACAAGGAGGTCTACAATCAACAG
*SfUGT34*-R	CCTATGAGTGGAACGCCTACTA
*SfUGT40*-F	CTACAGTCCACAGATGAGGCTAT
*SfUGT40*-R	TCTTCCACAGTCAACTTCTCCATA
*SfUGT48*-F	GCTGGTTCCTTGCTCATACTG
*SfUGT48*-R	AACTGCTCCACAACAATCACAT

Note: These seven *UGT* genes were selected based on their relatively high expression levels, broad phylogenetic distribution, and potential tissue- or stage-specific expression patterns, making them representative and suitable for further functional analysis.

**Table 2 insects-16-00601-t002:** *SfUGT* gene replication and Ka/Ks value analysis in *S. frugiperda*.

Duplicated Genes	Ka	Ks	Ka/Ks	Divergence Time (Million Years)	Duplicated Type
*SfUGT1-SfUGT5*	0.40192179	1.76697972	0.22746259	13.6	Tandem replication
*SfUGT2-SfUGT4*	0.37795081	3.3103476	0.11417254	25.5	Tandem replication
*SfUGT10-SfUGT12*	0.27759684	NaN	NaN		Tandem replication
*SfUGT11-SfUGT13*	0.27068699	NaN	NaN		Tandem replication
*SfUGT16-SfUGT17*	0.1520578	0.91537335	0.1661156	7	Tandem replication
*SfUGT18-SfUGT16*	0.15772597	0.85685295	0.18407589	6.6	Tandem replication
*SfUGT17-SfUGT19*	0.10253229	0.47379352	0.21640712	3.6	Tandem replication
*SfUGT20-SfUGT27*	0.35321099	4.72283199	0.07478796	36.3	Tandem replication
*SfUGT21-SfUGT26*	0.09656067	0.40050911	0.24109481	3.1	Tandem replication
*SfUGT22-SfUGT28*	0.13243781	0.49979133	0.2649862	3.8	Tandem replication
*SfUGT24-SfUGT25*	0.22994111	1.43336234	0.16042079	11	Tandem replication
*SfUGT26-SfUGT27*	0.14655933	0.53908785	0.27186538	4.1	Tandem replication
*SfUGT34-SfUGT33*	0.21169376	1.09658972	0.19304737	8.4	Tandem replication
*SfUGT34-SfUGT35*	0.21264377	1.03035232	0.20637967	7.9	Tandem replication
*SfUGT36-SfUGT37*	0.07379752	0.23438925	0.3148503	1.8	Tandem replication
*SfUGT37-SfUGT38*	0.01976299	0.1246974	0.15848757	1	Tandem replication
*SfUGT40-SfUGT43*	0.12685392	0.57297678	0.22139452	4.4	Tandem replication
*SfUGT41-SfUGT42*	0.31637091	1.07271261	0.29492606	8.3	Tandem replication
*SfUGT41-SfUGT44*	0.30201086	1.72279897	0.17530244	13.3	Tandem replication
*SfUGT43-SfUGT46*	0.18612883	0.98093876	0.18974562	7.5	Tandem replication
*SfUGT44-SfUGT45*	0.28981858	1.44860626	0.20006719	11.1	Tandem replication
*SfUGT45-SfUGT46*	0.18289155	1.01046947	0.18099661	7.8	Tandem replication
*SfUGT47-SfUGT48*	0.58233839	2.35592325	0.24718054	18.1	Tandem replication

Note: Ka/Ks is unitless. Divergence time is expressed in million years ago (Mya), calculated using the formula T = Ks/(2λ), where λ = 6.5 × 10^−9^.

**Table 3 insects-16-00601-t003:** Analysis of subcellular localization and physicochemical characteristics of the *SfUGT* gene family in *S. frugiperda*.

Gene Name	Gene ID	Number of Amino Acid	Molecular Weight	Theoretical pI	Instability Index	Aliphatic	Grand Average of Hydropathicity	Subcellular Localization
*SfUGT1*	118265011	342	38,957.63	9.16	47.02	107.43	0.111	mito
*SfUGT2*	118265039	373	41,734.13	8.88	44.43	99.57	−0.005	extr
*SfUGT3*	118265260	534	60,369.81	7.98	53.91	104.96	0.16	plas
*SfUGT4*	118265297	526	59,061.97	7.78	40.81	108.42	0.134	plas
*SfUGT5*	118265425	526	59,083.03	8.25	39.12	107.87	0.14	plas
*SfUGT6*	118266879	530	58,835.84	7.82	40.64	103.98	0.053	extr
*SfUGT7*	118267687	544	62,128.39	9.07	57.92	95.18	−0.042	plas
*SfUGT8*	118267688	544	61,991.21	9.01	56.57	95.9	−0.046	plas
*SfUGT9*	118268440	520	59,548.25	6.77	48.3	108.69	−0.024	E.R.
*SfUGT10*	118269227	525	60,827.42	6.77	47.21	100.44	−0.108	E.R.
*SfUGT11*	118269255	525	60,668.23	6.48	45.96	103.96	−0.069	E.R.
*SfUGT12*	118269333	521	60,644.21	8.13	44.15	96.14	−0.181	E.R.
*SfUGT13*	118269358	521	60,690.24	8.13	43.91	95.39	−0.189	E.R.
*SfUGT14*	118275940	539	61,284.8	9.37	40.77	102.02	0.004	E.R.
*SfUGT15*	118277768	523	59,649.65	8.76	39.57	102.89	−0.038	nucl
*SfUGT16*	118277895	522	59,740.24	7	39.78	101.76	−0.077	plas
*SfUGT17*	118277896	521	60,228.23	9.15	37.79	99.5	−0.14	E.R.
*SfUGT18*	118277897	521	59,245.87	8.75	38.97	108.89	0.024	plas
*SfUGT19*	118277898	528	60,765.61	8.52	41.42	104.66	−0.096	E.R.
*SfUGT20*	118277899	524	59,178.42	8.87	42.89	100.06	0.029	E.R.
*SfUGT21*	118277900	523	59,652.4	8.92	41.9	101.72	−0.156	E.R.
*SfUGT22*	118277901	523	59,365.08	8.33	32.63	98.91	−0.002	E.R.
*SfUGT23*	118277902	522	59,149.35	8.89	45.66	97.28	0.001	plas
*SfUGT24*	118277903	522	59,049.7	6.81	40.21	102.49	0.014	E.R.
*SfUGT25*	118277904	522	59,434.31	8.05	40.88	99.48	−0.075	E.R.
*SfUGT26*	118277905	520	59,257.15	9.05	41.72	104.73	−0.084	plas
*SfUGT27*	118277906	438	50,218.57	9.06	46.38	96.96	−0.118	pero
*SfUGT28*	118278039	1102	124,725.23	8.24	35.96	107.22	0.016	E.R.
*SfUGT29*	118278122	508	58,469.71	5.89	55.23	107.24	−0.089	plas
*SfUGT30*	118278231	236	26,641.16	9.35	33.12	101.14	−0.1	cyto
*SfUGT31*	118278649	514	58,243.33	8.92	56.53	105.47	0.047	plas
*SfUGT32*	118278849	520	57,729.49	9.26	47.01	98.85	0.018	pero
*SfUGT33*	118279153	518	59,256.76	8.26	44.09	103.05	−0.116	E.R.
*SfUGT34*	118279154	516	59,179.94	8.77	43.66	100.23	−0.151	E.R.
*SfUGT35*	118279155	515	58,893.45	7.27	47.61	99.32	−0.139	E.R.
*SfUGT36*	118279189	518	58,821.86	8.63	37.54	106.66	−0.025	E.R.
*SfUGT37*	118279190	322	37,089.67	8.29	36.55	105.62	0.003	extr
*SfUGT38*	118279191	286	32,873.72	6.55	36.41	109.34	0.134	E.R.
*SfUGT39*	118279357	512	58,603.17	6.32	42	106.58	−0.026	E.R.
*SfUGT40*	118279411	525	59,503.47	8.22	41.82	105.24	0.011	E.R.
*SfUGT41*	118279412	522	59,613.42	8.34	43.32	102.3	−0.056	E.R.
*SfUGT42*	118279413	521	58,861.73	8.83	50.4	105.47	−0.012	plas
*SfUGT43*	118279415	521	59,225.95	8.48	40.83	103.63	−0.016	E.R.
*SfUGT44*	118279416	521	59,580.05	6.61	50.23	106.07	−0.088	plas
*SfUGT45*	118279417	520	59,555.16	7.81	41.16	106.27	−0.067	E.R.
*SfUGT46*	118279418	520	59,370.96	7.79	47.38	109.42	−0.089	E.R.
*SfUGT47*	118281901	516	58,246.87	8.92	42.41	104.81	0.12	plas
*SfUGT48*	118281915	1101	125,967.94	9.11	44.63	95.44	−0.099	E.R.

Note: E.R.: endoplasmic reticulum; nucl: nucleus; extr: extracellular space; plas: plasma membrane; pero: peroxisome; cyto: cytoplasm; mito: mitochondria.

**Table 4 insects-16-00601-t004:** Examination of the secondary structure and signaling peptide of the SfUGT protein in *S. frugiperda*.

Gene Name	Alpha Helix	Beta Turn	Random Coil	Extended Strand
*SfUGT1*	141 (41.23%)	9 (2.63%)	137 (40.06%)	55 (16.08%)
*SfUGT2*	162 (43.43%)	17 (4.56%)	140 (37.53%)	54 (14.48%)
*SfUGT3*	253 (47.38%)	28 (5.24%)	180 (33.71%)	73 (13.67%)
*SfUGT4*	251 (47.72%)	26 (4.94%)	179 (34.03%)	70 (13.31%)
*SfUGT5*	242 (46.01%)	22 (4.18%)	194 (36.88%)	68 (12.93%)
*SfUGT6*	259 (48.87%)	25 (4.72%)	181 (34.15%)	65 (12.26%)
*SfUGT7*	251 (46.14%)	27 (4.96%)	191 (35.11%)	75 (13.79%)
*SfUGT8*	261 (47.98%)	20 (3.68%)	192 (35.29%)	71 (13.05%)
*SfUGT9*	232 (44.62%)	23 (4.42%)	192 (36.92%)	73 (14.04%)
*SfUGT10*	255 (48.57%)	21 (4.00%)	181 (34.48%)	68 (12.95%)
*SfUGT11*	244 (46.48%)	25 (4.76%)	179 (34.10%)	77 (14.67%)
*SfUGT12*	244 (46.83%)	22 (4.22%)	181 (34.74%)	74 (14.20%)
*SfUGT13*	239 (45.87%)	25 (4.80%)	185 (35.51%)	72 (13.82%)
*SfUGT14*	243 (45.08%)	29 (5.38%)	191 (35.44%)	76 (14.10%)
*SfUGT15*	239 (45.70%)	22 (4.21%)	192 (36.71%)	70 (13.38%)
*SfUGT16*	239 (45.79%)	30 (5.75%)	184 (35.25%)	69 (13.22%)
*SfUGT17*	241 (46.26%)	21 (4.03%)	187 (35.89%)	72 (13.82%)
*SfUGT18*	233 (44.72%)	24 (4.61%)	186 (35.70%)	78 (14.97%)
*SfUGT19*	248 (46.97%)	24 (4.55%)	184 (34.85%)	72 (13.64%)
*SfUGT20*	249 (47.52%)	25 (4.77%)	180 (34.35%)	70 (13.36%)
*SfUGT21*	243 (46.46%)	28 (5.35%)	175 (33.46%)	77 (14.72%)
*SfUGT22*	233 (44.55%)	23 (4.40%)	189 (36.14%)	78 (14.91%)
*SfUGT23*	248 (47.51%)	23 (4.41%)	184 (35.25%)	67 (12.84%)
*SfUGT24*	234 (44.83%)	23 (4.41%)	191 (36.59%)	74 (14.18%)
*SfUGT25*	235 (45.02%)	24 (4.60%)	194 (37.16%)	69 (13.22%)
*SfUGT26*	240 (46.15%)	22 (4.23%)	189 (36.35%)	69 (13.27%)
*SfUGT27*	200 (45.66%)	20 (4.57%)	165 (37.67%)	53 (12.10%)
*SfUGT28*	446 (40.47%)	90 (8.17%)	338 (30.67%)	228 (20.69%)
*SfUGT29*	250 (49.21%)	22 (4.33%)	165 (32.48%)	71 (13.98%)
*SfUGT30*	104 (44.07%)	17 (7.20%)	83 (35.17%)	32 (13.56%)
*SfUGT31*	240 (46.69%)	26 (5.06%)	176 (34.24%)	72 (14.01%)
*SfUGT32*	230 (44.23%)	23 (4.42%)	194 (37.31%)	73 (14.04%)
*SfUGT33*	231 (44.59%)	22 (4.25%)	193 (37.26%)	72 (13.90%)
*SfUGT34*	236 (45.74%)	26 (5.04%)	181 (35.08%)	73 (14.15%)
*SfUGT35*	246 (47.77%)	21 (4.08%)	179 (34.76%)	69 (13.40%)
*SfUGT36*	245 (47.30%)	24 (4.63%)	178 (34.36%)	71 (13.71%)
*SfUGT37*	157 (48.76%)	11 (3.42%)	111 (34.47%)	43 (13.35%)
*SfUGT38*	124 (43.36%)	8 (2.80%)	104 (36.36%)	50 (17.48%)
*SfUGT39*	234 (45.70%)	24 (4.69%)	177 (34.57%)	77 (15.04%)
*SfUGT40*	231 (44.00%)	23 (4.38%)	198 (37.71%)	73 (13.90%)
*SfUGT41*	241 (46.17%)	26 (4.98%)	179 (34.29%)	76 (14.56%)
*SfUGT42*	236 (45.30%)	25 (4.80%)	182 (34.93%)	78 (14.97%)
*SfUGT43*	233 (44.72%)	22 (4.22%)	189 (36.28%)	77 (14.78%)
*SfUGT44*	244 (46.83%)	28 (5.37%)	181 (34.74%)	68 (13.05%)
*SfUGT45*	231 (44.42%)	23 (4.42%)	191 (36.73%)	75 (14.42%)
*SfUGT46*	240 (46.15%)	23 (4.42%)	189 (36.35%)	68 (13.08%)
*SfUGT47*	231 (44.77%)	20 (3.88%)	189 (36.63%)	76 (14.73%)
*SfUGT48*	474 (43.05%)	55 (5.00%)	391 (35.51%)	181 (16.44%)

## Data Availability

The original contributions presented in this study are included in the article. Further inquiries can be directed to the corresponding authors.

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
