# Peer review of "Comprehensive Analysis of the UGT Gene Superfamily in Spodoptera frugiperda"

_insects, 2025, doi:10.3390/insects16060601_

Round 1
Reviewer 1 Report
Comments and Suggestions for Authors
Wu et al. give an excellent bioinformatic and expression‐based depiction of the UDP-glucuronosyl transferase (UGT) superfamily in Spodoptera frugiperda, 48 genes of which are identified, their chromosomal distribution determined, and their expression profiled during developmental stages and tissues using transcriptome and qPCR data. The strengths lie in the scope of analyses—gene duplication, physicochemical properties, secondary structure, phylogenetics among species, and tissue/developmental expression—and the validation of key genes through qPCR. However, the study heavily relies on in silico prediction with few functional tests, and some methodological data are very poorly explained. Over-all quality: 80/100. Quality of language: 8/10.
The Methods section is overall explained but without necessary information on setting parameters, statistics, and quality control. The homolog searching criteria ("Liu Yang's method") and TBtools settings, for example, are not explained fully and thus hard to reproduce. The phylogenetic reconstruction lacks information on alignment trimming, model selection, and support node values. For the qPCR analyses, primer efficiency testing and validation of normalization beyond GAPDH use is not mentioned, nor are statistical tests (e.g., the manner in which differences were tested). The Discussion puts the results in good context for insect detoxification research but overinterprets correlative expression patterns as functional roles in the absence of experimental evidence; it also presents duplicate results rather than combining them into mechanistic insights or directions for future work.
Critical Points:
The homolog identification pipeline ("Liu Yang's method") is not described well enough—parameter thresholds, domain detection criteria, and false positive controls are lacking, undermining confidence in the gene set.
Phylogenetic analysis does not give alignment quality scores, evolutionary model choice, or bootstrap/SH-aLRT support values, which undermine the evolutionary inferences on subfamily classification and orthology.
Ka/Ks ratios and divergence time estimates are presented without confidence intervals or rate heterogeneity tests; two NaN values are noted but not interpreted biologically or methodologically.
qPCR protocol lacks information on primer specificity and efficiency, reference gene stability between samples, and statistical tests used to assess significance, which undermines validity of expression claims.
Integration of transcriptomic TPM values and qPCR fold changes is not fully concordant (e.g., SfUGT30 high in adults by qPCR but not transcriptome), yet this is not addressed or resolved.
Discussion comments sometimes equate high expression with functional detoxification activities without biochemical or genetic assays to define causality and overemphasize the roles of individual UGTs.
Chromosomal mapping figures and motif analyses are presented but not compared quantitatively (e.g., motif conservation scores), preventing insight into evolutionary constraint.
The conclusion mentions pesticide development potential but fails to link individual UGT targets to pesticide response assays, making this recommendation speculative.
Minor Points
Legend in tables lacks unit for Ka/Ks ratio and divergence time, requiring reader inference.
Some gene names (e.g., SfUGT16, SfUGT21) are inconsistently italicized in text.
Labels (e.g., "plas," "E.R.") in figures are not defined in captions but rather are defined only within tables, leaving transient confusion.
The Materials and Methods section of the manuscript duplicatively states software versions and URLs; these should be tabulated in brief format for readability.
The occasional grammatical errors, like "only had only 1 gene" (p. 2), must be corrected.
The phrase "functional roles" is used generally—whether roles are meant to be detoxification, development, or metabolism would make it more precise.
Impressions of Results: The multi-level analyses firmly rulebook the UGT repertoire of S. frugiperda and show divergent expression patterns suggesting developmental and tissue-specific functions; however, in the absence of supporting biochemical or genetic evidence, the significance for pest management is provisional.
Recommendation to Editor: Recommend major revision to clarify methodological transparency, increase statistical rigor, and temper functional claims until experimental verification.
Reviewer 2 Report
Comments and Suggestions for Authors
This manuscript reports a genome-wide identification and analysis of the UGT gene superfamily in Spodoptera frugiperda, covering gene structure, chromosomal location, expression patterns, and phylogenetic relationships. The general approach is valid and the topic is relevant, but there are several aspects that need to be improved for the work to be more solid and informative.
One of my main concerns is the lack of clarity about what’s new in this study. Similar analyses have already been published. Ahn et al. (2012) and Israni et al. (2020), for example, covered UGT gene families in insects, including S. frugiperda. So I’d like to ask what is this study adding? Are you identifying UGTs that weren’t previously reported? Are there differences in gene duplication patterns or expression profiles worth highlighting? Right now, this part is not clear.
The study is based entirely on in silico analyses and qPCR expression data, which is fine for a gene family survey. But many of the statements about gene function read as if those roles were already confirmed. It’s okay to propose functions based on homology or expression patterns, but these should be stated as hypotheses. Phrases like “may be involved in” or “possibly related to” would make the interpretation more balanced. Otherwise, it feels too speculative.
Methods
Some concerns about the specimens of S. frugiperda used in the study. It says the insects came from maize fields and were reared for “successive generations,” but it doesn’t say when the collection was done, what the field conditions were, or how many generations were kept in the lab. The number of generations maintained under laboratory conditions should be explicitly stated. Long-term laboratory rearing can lead to significant physiological and gene expression changes, especially genes related to detoxification, metabolism, and stress responses. Without clear context, it is difficult to interpret whether the expression profiles reflect natural physiological states or are artifacts of lab adaptation.
The statistical methods for the qPCR data aren’t clearly described. How many replicates were done? What statistical test was used to compare expression levels? That needs to be added too.
Results and Discussion
There’s a lot of repetition between the results and the discussion. In many places, the same expression patterns are restated almost word for word. That section would be much easier to follow if the overlapping content were reduced or reorganized.
The discussion is too short and doesn’t go into much depth. It mostly summarizes what was already described in the results, without really connecting the findings to other studies or discussing the broader implications. There’s no mention of how this information could be useful for pest control, or any discussion of the study’s limitations.
Other comments
The gene names (SfUGT) are not used consistently. Please standardize them throughout the manuscript and include gene IDs when necessary.
Some phrases could be clearer. For example, “substantial activity in amino acid sequences” (line 339) isn’t meaningful as written. Also, “proteins may help eliminate reactive oxygen species” (line 210) is a strong statement and needs support if it’s going to be included.
Figures 2 to 5 are hard to interpret. The phylogenetic trees don’t include bootstrap values, and the figure legends are missing important details. These figures must be revised to improve readability.
Comments on the Quality of English Language
The English is generally understandable but needs revision for clarity and flow.
Reviewer 3 Report
Comments and Suggestions for Authors
In this manuscript bioinformatics was used to analyze the UGT gene family in a fall armyworm, Spodoptera frugiperda. A total of 48 UGT genes were identified from the genome of S. frugiperda and found to be distributed across 10 different chromosomes. The tissue-specific expression and functional analysis of these UGT genes provide in-depth insights into the structural characteristics of the insect UGT gene family. However, this reviewer has three major concerns:
- Nomenclature of insect UGTs: Insect UGTs predominantly use UDP-glucose as the sugar donor, and their enzymatic activity assays are conducted with UDP-glucose. Therefore, the full name of insect UGTs should be UDP-glycosyltransferase, as used in most literatures, rather than UDP-glucuronosyltransferases. The latter term refers specifically to mammalian UGTs, which utilize UDP-glucuronic acid as the sugar donor (see: Bock KW. The UDP-glycosyltransferase (UGT) superfamily expressed in humans, insects and plants: Animal-plant arms-race and co-evolution. Biochem Pharmacol 2016, 99:11-17; Ahn et al. Comparative analysis of the UDP-glycosyltransferase multigene family in insects. Insect Biochem Molec Biol 2012, 42:133-147).
- Numbering convention for insect UGT genes: According to the UGT family nomenclature rules, UGT1-8 are assigned to vertebrates, while UGT31-50 are designated for insects (Bock 2016; Ahn et al., 2012). This convention has been consistently followed in naming UGTs in Drosophila, Aedes aegypti, Anopheles gambiae, Helicoverpa armigera, Spodoptera exigua, Bombyx mori, and Tribolium castaneum. Previous studies on S. frugiperda UGTs also adhered to this rule. However, in this study, the 48 identified UGT genes were numbered based on their chromosomal order (Fig 1), which could introduce confusion and hinder systematic understanding of insect UGT structure and function. As a comprehensive study on the insect UGT family, the authors should adopt the internationally accepted nomenclature for S. frugiperda UGT genes.
- Functional domain analysis is lacking: While the paper claims to be a "comprehensive analysis" of the UGT gene family, it primarily focuses on gene structure and secondary protein structure prediction, omitting critical functional domain analysis. Given the availability of advanced bioinformatics tools (e.g., AlphaFold) and extensive prior research on insect UGT functions, the authors should leverage these resources to provide deeper functional insights.
Minor comments:
- Line 75: sulfonamide should be corrected to sulfoxaflor.
- The study analyzed the developmental stage- and tissue-specific expression of only 7 UGT genes out of 48. What was the rationale for selecting these specific genes? A more comprehensive analysis of all UGT genes would strengthen the study’s conclusions.
Round 2
Reviewer 1 Report
Comments and Suggestions for Authors
The authors have responded to most of my comments, and the manuscript is much clearer, methodologically stronger, and with better interpretation of data. The most prominent changes include elaborate descriptions of the bioinformatics workflow, parameters in phylogenetic analysis, qPCR validation methods, and uniform formatting throughout. Yet, there are some points that have to be dealt with in the final revised version:
- Clearly mention the limitations regarding the absence of confidence intervals for Ka/Ks ratios and the absence of quantitative motif conservation analysis (Comments 3 and 7);
- Briefly discuss the methodological discrepancies between qPCR and transcriptomic data in Results or Discussion (Comment 5); and
- Incorporate all grammar corrections and do a final full proofreading.
These small changes will increase transparency and bring the manuscript up to the standards of the journal.
Reviewer 2 Report
Comments and Suggestions for Authors
Thank you for the revision. This version of the manuscript shows clear improvements. The introduction now provides a stronger context and clearly explains how the study builds upon previous work. The discussion includes more relevant references and better integrates the findings with potential applications.
Here are a few additional suggestions:
In Figure 2, the grouping of clades remains difficult to interpret. Use color shading or branch coloring to clearly separate Groups A, B, and C. Also, include bootstrap or other support values directly on the branches to strengthen the presentation.
In Figure 3, the species are well distinguished by color, but the tree still lacks support values. Add bootstrap or SH-aLRT values to improve the credibility of the phylogenetic relationships shown.
Some sentences still read awkwardly, despite the overall improvement in English. Review the text again to eliminate literal translations and improve fluency.
Mention the absence of statistical testing for the qPCR results again in the Discussion or Conclusions, to clearly acknowledge the limitation.
Comments on the Quality of English LanguageThe manuscript shows noticeable improvement in language use compared to the original version. Most sections now read more clearly, and the speculative language has been appropriately moderated. However, several sentences still contain awkward phrasing or overly literal translations that affect fluency. A final review by a native English speaker or professional editor is recommended to improve flow and clarity throughout the text.
Reviewer 3 Report
Comments and Suggestions for Authors
I have no other comments
